# Impact of Changing Surgical Strategies on Clinical Outcomes in Patients with Parotid Carcinoma: A 53-Year Single-Institution Experience

**DOI:** 10.3390/medicina57080745

**Published:** 2021-07-23

**Authors:** Hirotaka Yamamoto, Tsuyoshi Kojima, Yusuke Okanoue, Shuya Otsuki, Koki Hasebe, Ryohei Yuki, Ryusuke Hori

**Affiliations:** 1Department of Otolaryngology, Tenri Hospital, 200 Mishima-cho, Tenri, Nara 632-8552, Japan; h_yamamoto@ent.kuhp.kyoto-u.ac.jp (H.Y.); t_kojima@ent.kuhp.kyoto-u.ac.jp (T.K.); yusuke.okanoue@gmail.com (Y.O.); s_otsuki@ent.kuhp.kyoto-u.ac.jp (S.O.); k_hasebe@ent.kuhp.kyoto-u.ac.jp (K.H.); r_yuki@ent.kuhp.kyoto-u.ac.jp (R.Y.); 2Department of Otolaryngology, Shizuoka City Hospital, 10-93 Otte-cho, Aoi-ku, Shizuoka 420-8630, Japan; 3Department of Otolaryngology—Head and Neck Surgery, School of Medicine, Fujita Health University, 1-98 Dengakugakubo, Kutsukake-cho, Toyoake 470-1192, Aichi, Japan

**Keywords:** parotid carcinoma, partial parotidectomy, facial nerve palsy, prognostic factors, neck dissection

## Abstract

*Background and Objectives*: We investigated the clinical outcomes of patients who underwent surgery for parotid carcinoma in a single institution during a 53-year period. This study aimed to estimate the impact of changing the surgical approach to parotid carcinoma on clinical outcomes including the incidence rate of the facial nerve palsy. *Materials and Methods*: Sixty-seven patients with parotid carcinoma who underwent surgery between 1966 and 2018 were retrospectively reviewed. Group A consisted of 29 patients who underwent surgery from 1966 to 2002, and Group B consisted of 38 patients from 2002 to 2018. Treatment outcomes were estimated. Additionally, candidate prognostic factors of Group B, the current surgical approach group, were evaluated. *Results*: Partial parotidectomy and total parotidectomy were performed in 35 and 32 patients, respectively. Partial parotidectomy was performed in 4 patients in Group A and 31 patients in Group B, with a predominant increase in Group B. The facial nerve was preserved in 43 patients, among whom 8 in Group A (8/17; 47.1%) and 7 in Group B (7/26; 26.9%) had temporary postoperative facial nerve palsy. Postoperative radiotherapy was performed on 35 patients. The 5-year OS, DSS, and DFS rates for Group A were 77.1%, 79.9%, and 71.5%, respectively. The 5-year OS, DSS, and DFS rates for Group B were 77.1%, 77.1%, and 72.4%, respectively. Clinical T4 stage, clinical N+ stage, stage IV disease, and tumor invasion of the facial nerve were independent prognostic factors in Group B. *Conclusions*: The incidence of facial nerve palsy in the current surgical approach group decreased compared with that in the previous surgical approach group. The current surgical management and treatment policies for parotid carcinoma have led to improved outcomes.

## 1. Introduction

Parotid carcinoma is a relatively rare disease that accounts for less than 5% of head and neck malignancies worldwide [1]. In Japan, the annual incidence of parotid carcinoma is approximately 3 out of every 1,000,000 people [2]. The World Health Organization (WHO) recognizes at least 24 histological types of salivary gland carcinomas [3]. Their rarity and histological variety have prevented the development of a definitive surgical policy and prognostic prediction method for this disease. Treatment strategies for parotid carcinoma include surgical resection, radiotherapy, and chemotherapy. Surgical resection is the primary treatment for operable parotid carcinoma; in addition, postoperative radiotherapy is used to reduce the incidence of local and cervical lymph node recurrence in high-risk patients [4]. However, the institutional policy and surgeon’s preference are the main determinants of the surgical method used for treating parotid carcinoma, area of resection of the parotid gland, indications for neck dissection in clinical N0 (cN0) cases, and treatment of the facial nerve.

In the past, total parotidectomy was the standard treatment of choice for parotid gland carcinoma with a poor prognosis [5]. Total parotidectomy involves the unnecessary removal of normal parotid tissue, which may result in postoperative impairment of the salivary glands and facial nerve function [6]. Therefore, partial parotidectomy has emerged as a more conservative approach over the past 2 decades [7,8]. In partial parotidectomy, only the tumor-bearing area of the parotid parenchyma is removed. The main trunk of the facial nerve is identified, and the facial nerve branch adjacent to the tumor is dissected and preserved. Compared to total parotidectomy, partial parotidectomy is associated with a low incidence of surgical complications, such as facial palsy, loss of sensation of the ear, poor aesthetics, and Frey’s syndrome [9]. Total parotidectomy is not necessary for low-grade cancers if the tumor is completely resectable, as alternative approaches can be used [6,10]. In terms of high-grade cancers, some reports have uniformly recommended total parotid resection [11], while others have suggested that lobectomy is sufficient for T1 and T2 disease without facial nerve involvement [6,12]. The indications for prophylactic neck dissection for cN0 parotid carcinoma are unclear. While several authors recommend prophylactic neck dissection in all patients with parotid carcinoma, others recommend postoperative radiotherapy or observation [13,14,15,16].

At our institute, the surgical strategy for parotid carcinoma shifted from total parotidectomy to partial parotidectomy in 2002. Partial parotidectomy was preferred because it preserves the facial nerve as much as possible in patients without preoperative facial palsy. In addition, the neck management for cN0 parotid carcinoma was changed from prophylactic neck dissection at cT3 and T4 cases before 2002 to no prophylactic neck dissection for cases after 2002. These changes in the surgical management and treatment policies may have improved treatment outcomes, such as postoperative facial nerve function. This study aimed to assess the changes in clinical outcomes of patients treated for parotid carcinoma at our institution due to changes in surgical strategy before/after 2002. To the best of our knowledge, this is the first study to compare postoperative outcomes before and after implementing specific surgical management policy changes for parotid carcinoma at a single institution.

## 2. Materials and Methods

We retrospectively analyzed 67 patients with primary parotid carcinoma who underwent surgical treatment at the Tenri Hospital between January 1966 and December 2018. Patients who received palliative treatment were excluded from this study. Patients were classified into two groups based on the date of surgery. Group A, the previous surgical approach group, consisted of 29 patients who underwent surgery from January 1966 to December 2002. During this period, total parotidectomy was chosen as the main surgical method, and prophylactic neck dissection was performed for cN0 patients with cT3 or T4. Group B, the current surgical approach group, consisted of 38 patients who underwent surgery from January 2002 to December 2018. In the current surgical approach group, the surgical technique was changed to a conservative approach, such as partial parotidectomy or no prophylactic neck dissection in cN0 cases with cT3 or T4. All patients received a preoperative physical examination, ultrasonography (US), contrast-enhanced head and neck computed tomography (CT), and/or magnetic resonance imaging (MRI). CT and/or MRI were performed in order to find the location, extent, and invasion of the primary tumor as well as the presence of nodal metastasis. Ultrasound-guided fine-needle aspiration biopsy was performed for the primary tumor and for lymph nodes suspected to be metastatic by the imaging studies. In cases of suspected distant metastasis, bone scintigraphy or FDG-PET was conducted. 

The personal records of all patients were reviewed, and age, sex, clinical, surgical, pathological, and follow-up data were extracted. The clinical stage was classified based on the 8th edition of the tumor–node–metastasis (TNM) classification [17]. The histological type was classified based on the 4th edition of the WHO classification [3]. Treatment outcomes of the two groups including overall survival (OS), disease-specific survival (DSS), disease-free survival (DFS), and the incidence rate of the facial nerve palsy were examined. The following candidate prognostic factors were assessed: age, sex, cT stage, cN stage, intraoperative facial nerve invasion, histological grade, and postoperative radiotherapy. We analyzed local, cervical, and distant recurrences in all patients; we also analyzed cervical lymph node recurrence in cN0 patients and recurrences in patients treated with postoperative radiotherapy.

In terms of the surgical management and treatment policy for parotid carcinoma, total parotidectomy was chosen as the main surgical method in Group A. On the other hands, partial parotidectomy was chosen as the main surgical method in Group B, and total parotidectomy was chosen in cases in which almost all the nerves were involved in a large tumor. The facial nerve was resected in patients with preoperative facial nerve palsy in both groups. In patients without preoperative facial nerve palsy, if it was found to be in contact with the tumor intraoperatively, the nerve was sacrificed in group A, while the nerve was detached from the tumor and preserved in group B. Postoperative facial nerve function was evaluated on the first postoperative day by clinical examination and graded according to the Yanagihara 40-point system (Yanagihara score) [18]. Until 2002, prophylactic neck dissection was performed in patients with advanced parotid carcinomas (clinical T3 or T4), even in the cN0 cases. Since 2002, prophylactic neck dissection was not performed in patients with cN0 parotid carcinoma. In addition, postoperative radiotherapy was administered to patients with high-grade malignancy, T4 tumor, perineural or lympho-vascular invasion, positive resection margin on histopathological evaluation, or multiple cervical lymph node metastases. In patients without cervical lymph node metastasis, the irradiation field (50 Gy) was limited to the parotid gland. In patients with cervical lymph node metastasis, the irradiation field (50 Gy) was extended to the lateral neck, and chemoradiation therapy was performed.

Group A patients were followed up until the time of death or until 2002, whereas Group B patients were followed up until the time of death or until February 2020. US, CT, and/or MRI were performed every 3 months during the first year and every 6 months in the subsequent years for the evaluation of regional and cervical lymph nodes and distant metastases.

Categorical variables were compared using the chi-square test, and continuous variables were compared using the Student’s t-test. The treatment outcomes, including OS, DSS, and DFS, were evaluated using the Kaplan–Meier method, and groups were compared using the log-rank test. A *p*-value < 0.05 was considered statistically significant. All statistical analyses were performed with EZR (Saitama Medical Centre, Jichi Medical University), which is a graphical user interface for R (The R Foundation for Statistical Computing, Vienna, Austria, version 2.13.0) [19].

## 3. Results

### 3.1. Patient Characteristics

The patient characteristics of the two groups, including age, sex, cT and cN classifications, disease stage, surgical methods, and preoperative/postoperative facial nerve palsy, postoperative radiotherapy, and recurrence rate are presented in Table 1. No statistically significant differences in sex, cT and cN classification, disease stage, facial nerve resection, preoperative/postoperative facial nerve palsy, and postoperative radiotherapy were noted between the two groups. Group B had a significantly higher median age. Partial parotidectomy was performed in 4 patients in Group A, and 31 patients in Group B, with a predominant increase in Group B. Prophylactic neck dissection was performed in 5 patients in Group A (2 on type modified radical, and 3 on type selective), among which only 1 patient had occult cervical neck metastasis. Therapeutic neck dissection was performed in 7 patients in Group A (3 on type radical, and 4 on type modified radical) and 2 patients in Group B (both on type selective) with clinically positive nodes.

Regarding facial nerve function and surgical treatment for facial nerve, 3 patients in Group A and 1 patient in Group B exhibited preoperative facial palsy, and the facial nerve of these patients was sacrificed. There were 26 patients and 37 patients without preoperative facial nerve palsy in Group A and B, respectively. Of the 26 patients without preoperative facial nerve palsy in Group A, 9 sacrificed the facial nerve and 17 preserved the facial nerve during surgery. Of the 37 patients without preoperative facial nerve palsy in Group B, 11 sacrificed the facial nerve and 26 patients preserved the facial nerve during surgery. Among the 17 patients with facial nerve preservation in Group A, 8 (47.1%; 8/17) exhibited temporary postoperative facial nerve palsy (each Yanagihara score was less than 2 points), and among the 26 patients with facial nerve preservation in Group B, 7 (26.9%; 7/26) exhibited temporary postoperative facial nerve palsy. As a result, the incidence of postoperative facial nerve palsy in patients with preoperative facial palsy was 65.4% (17/26) in Group A and 48.6% (18/37) in Group B. Postoperative radiotherapy was performed in 35 patients, including 25 patients with high-grade carcinoma and 10 patients with low/intermediate-grade carcinoma (6 due to a T4 tumor and 3 due to perineural invasion).

### 3.2. Histopathologic Results

Histological grades and histologies in our study are shown in Table 2. Mucoepidermoid carcinoma was the most common histological type (34.3%), followed by acinic cell carcinoma (22.4%), carcinoma ex pleomorphic adenoma (17.9%), adenocystic carcinoma(9.0%), squamous cell carcinoma (6.0%), and others (10.4%). Thirty-three, 4, and 30 patients had low-grade, intermediate-grade, and high-grade carcinomas, respectively. Among the 23 patients with mucoepidermoid carcinoma, 14 had low-grade carcinomas, and 9 had high-grade carcinomas. Among the 12 patients with carcinoma ex pleomorphic adenoma, 1 had low-grade carcinomas, and 11 had high-grade carcinomas. Among the 6 patients with adenocystic carcinoma, 4 had intermediate-grade carcinomas, and 2 had high-grade carcinomas.

### 3.3. Treatment Outcomes

Median follow-up for patients was 61 months for the entire cohort, 58 months for Group A, and 65 months for Group B. The Kaplan–Meier curves of treatment outcomes of the two groups, including OS, DSS, and DFS, are shown in Figure 1. At the time of analysis, there were 17 deaths (25.3% of patients). The 5-year OS rates for Group B vs. Group A patients were 77.1% vs. 74.5%, respectively (*p* = 0.709) (Figure 1a). The 5-year DSS rates for Group B vs. Group A patients were 77.1% vs. 77.6%, respectively (*p* = 0.724) (Figure 1b). The 5-year DFS rates for Group B vs. Group A patients were 72.4% vs. 71.0%, respectively (*p* = 0.548) (Figure 1c). The results of the univariate analysis of clinical factors of Group A and Group B affecting the 5-year OS, DSS, and DFS rates are listed in Table 3. In Group A, candidate factors that showed a significant association with 5-year OS, DSS, and DFS included sex, cT classification, and disease stage. Clinical N classification was significantly associated with 5-year DSS and DFS, and histology grade was significantly associated only with 5-year DSS. In Group B, candidate factors that showed a significant association with 5-year OS, DSS, and DFS included cT classification, cN classification, disease stage, and facial nerve invasion. The Kaplan–Meier curves of Group B for OS stratified by these factors are shown in Figure 2. The 5-year OS rate was significantly higher in patients with cT1–3 disease than in those with cT4 disease (93.8% vs. 55.0%; *p* = 0.014; Figure 2a). The 5-year OS rate was significantly higher in clinically node-negative patients than in clinically node-positive patients (82.3% vs. 0%; *p* < 0.001; Figure 2b). The 5-year OS rate was significantly higher in patients with stage I–III disease than in those with stage IV disease (93.3% vs. 58.2%; *p* = 0.025; Figure 2c). Patients with intraoperative facial nerve invasion had a significantly lower 5-year OS rate than those without (50.0% vs. 89.2%; *p* = 0.009; Figure 2d). The remaining factors (age, sex, histological grade, and postoperative radiotherapy) were not significantly associated with the 5-year OS, DSS, or DFS; however, high-grade histology exhibited a trend for a poorer 5-year DSS (*p* = 0.102).

### 3.4. Recurrence

Details of the recurrent cases are summarized in Table 4. Group A cases in the first half of the study period had insufficient medical record review, and clinical outcomes were not fully known. Disease recurrence was observed in 17 patients (25.3%), including 11 patients with local recurrence (16.4%), 9 with cervical lymph node recurrence (13.4%), and 9 with distant metastasis (13.4%). Local recurrence was significantly lower in Group B than in Group A (1 in Group B vs. 10 in Group A, *p* < 0.01). From these recurrent cases, 10 deaths were noted. Histological diagnoses in patients with distant metastasis included mucoepidermoid carcinoma (4 cases), acinic cell carcinoma (2 cases), squamous cell carcinoma (1 case), adenocystic carcinoma (1 case), and carcinoma ex-pleomorphic adenoma (1 case). Cervical lymph node recurrence was observed in 5 of 58 cN0 cases (8.6%); these patients were managed with salvage surgery (in one case, the medical history could not be completed due to insufficient medical record research). Among the 35 patients who underwent postoperative radiotherapy, 9 experienced recurrences. Local recurrence and cervical lymph node recurrence were observed in 5 patients and 3 patients, respectively, all of whom underwent successful salvage surgery. Distant metastasis was observed in 5 patients; of these, 1 patient was successfully treated with salvage surgery.

## 4. Discussion

At our institute, total parotidectomy was performed for almost all parotid carcinomas until 2002. The facial nerve was resected during surgery in patients with preoperative facial nerve palsy or when the nerve was in contact with the tumor. Therefore, the incidence of postoperative facial nerve palsy was high until 2002: 47.1% (8/17) among patients with facial nerve preservation and 65.4% (17/26) among patients without preoperative facial nerve palsy. A few decades ago, the prevailing view was that more extensive resection at the expense of the facial nerve did not lead to improved outcomes [20], and the National Comprehensive Cancer Network guidelines recommended preservation of the facial nerve in the absence of facial nerve palsy, regardless of histological grade, unless the tumor had directly invaded and adhered to the nerve [21]. In 2002, we re-evaluated the surgical management of parotid carcinomas and identified partial parotidectomy as the primary surgical approach to reduce postoperative palsy incidence. If there was no preoperative facial palsy, the facial nerve was preserved by detaching it from the tumor, even in cases in which the nerve was in contact with the tumor. Thus, the incidence rate of facial nerve palsy in the current surgical strategy decreased to 26.9% (7/26) among patients with facial nerve preservation and 48.6% (18/37) among patients without preoperative facial nerve palsy, which was lower than that in the previous surgical strategy at our institute. Regarding local recurrence, the rate rather decreased from 35.7% (10/29) before 2002 to 2.6% (1/38) after 2002. Although the reasons are not known from the data of this study, it is possible that the skills of the surgeons have improved due to the increase in the number of head and neck cancer cases, including parotid cancer, since 2002. In addition, advances in imaging technology as well as radiation techniques (e.g., IMRT) might play an important role in staging of the cancers treated as well as the decrease of local recurrence rates. Compared to multicenter studies, single-center studies have less bias; this allowed us to compare the differences in treatment outcomes caused by changes in surgical management and treatment policies. To date, no studies have compared postoperative outcomes before and after the implementation of specific changes in the surgical management policy for parotid carcinoma at a single institution. Thus, this study provides insights into the potential impact that overarching policy changes can have on individual patient outcomes, such as the incidence of postoperative facial nerve palsy.

Until 2002, prophylactic neck dissection was performed in patients with advanced parotid carcinomas with clinical T3 or T4, even in the cN0 cases. Since 2002, prophylactic neck dissection has not been performed in patients with cN0 parotid carcinoma. Treatment policies for postoperative radiotherapy in cases of high-grade malignancy (or a tendency to infiltrate adjacent tissues, based on histological examination) remain unchanged. Until 2002, the 5-year OS rate after surgical treatment for parotid carcinoma was 74.5%. This is similar to the 5-year OS rate of 77.1% observed in the current surgical group. Some reports have suggested that prophylactic neck dissection should be considered in high-grade and T4 cases because of the high rate of occult lymph node metastasis [22]. Other studies have indicated that prophylactic neck dissection in cN0 cases does not improve survival and that the benefit of prophylactic neck dissection is limited in terms of prolonging survival [23,24,25]. In this study, only 1 of 5 cN0 patients had occult cervical lymph node metastasis in Group A, and 3 of the 36 cN0 patients showed late-onset cervical lymph node metastases in Group B, which were successfully managed by salvage neck dissection. These results suggest that prophylactic neck dissection is not always necessary and that early detection and prompt treatment of cervical lymph node metastases by means of strict postoperative follow-up are important.

In the latter period of this study, the facial nerve was preserved as much as possible if it could be detached from the tumor intraoperatively, even in cases of high-grade carcinoma. However, in such high-risk patients, the possibility of local recurrence and cervical lymph node recurrence should be considered. Many previous studies have reported a reduction in the incidence of postoperative local and cervical lymph node recurrences following postoperative radiotherapy [4], especially in cases involving T3–4 tumors, perineural invasion, and high-grade malignancy [26,27]. The median irradiation dose is reported to be 50 Gy [26,28]. In accordance with our institutional treatment policy, 35 patients with high-grade malignancy, T4 tumor, perineural invasion, positive margin on histopathological evaluation, or multiple lymph node metastases underwent postoperative adjunct radiotherapy in this study. Among the 35 patients, local recurrence was observed in 5 patients; cervical lymph node recurrence in 3 patients; and distant metastasis in 5 patients (including multiple forms of recurrence). Among patients receiving postoperative radiotherapy, local recurrence and/or cervical lymph node recurrence were observed in 3 patients who were managed with salvage surgery (in one case, the medical history could not be completed due to insufficient medical record research). Thus, postoperative radiotherapy appears to be effective in preventing local and cervical lymph node recurrence. 

The 5-year OS, DSS, and DFS rates for parotid carcinoma at our institute were comparable to those mentioned in other reports [29,30,31,32]. Univariate analysis suggested that cT4 stage, cN+ stage, stage IV disease, and facial nerve invasion were prognostic factors for parotid carcinoma. In terms of histological grade, in the first half of this study, the 5-year DSS rates for low-/intermediate-grade patients and high-grade patients were 94.4% and 34.3%, respectively, which were significantly different (*p* = 0.026), similar to those in other reports [23,33]. Although postoperative radiotherapy is effective in preventing local and cervical lymph node recurrences in patients with a poor prognosis, distant metastasis may ultimately determine the prognosis. Therefore, the development of novel and effective therapies is required; potential examples include molecular targeted therapy and immunotherapy, which have been shown to have antitumor activity against salivary gland carcinoma [34,35]. 

The present study has some limitations. These include its retrospective nature from a single-institution study. Furthermore, due to the rarity of parotid carcinoma, the number of patients per year is rather low. In addition, we were unable to make statistical comparisons among the different pathological types, owing to their large variety. In order to overcome these limitations, long-term multicenter studies with larger cohorts are required.

## 5. Conclusions

The incidence of postoperative facial nerve palsy among patients with facial nerve preservation in the current surgical strategy was lower than that documented in the previous surgical strategy at our institute; however, the 5-year each OS rate was similar between the two studies. cT4 stage, N+ stage, stage IV disease, and facial nerve infiltration were identified as independent prognostic factors. The current surgical management and treatment policies for parotid carcinoma have led to improved rates of postoperative facial nerve palsy as well as improved rates of OS, DSS, and DFS. Regardless of the effectiveness of postoperative radiotherapy in preventing local and cervical lymph node recurrences in cases of high-grade malignancy, distant metastasis ultimately determines the patients’ prognosis. Therefore, the development of new therapies, including molecular targeted therapy and immunotherapy, is warranted.

## Figures and Tables

**Figure 1 medicina-57-00745-f001:**
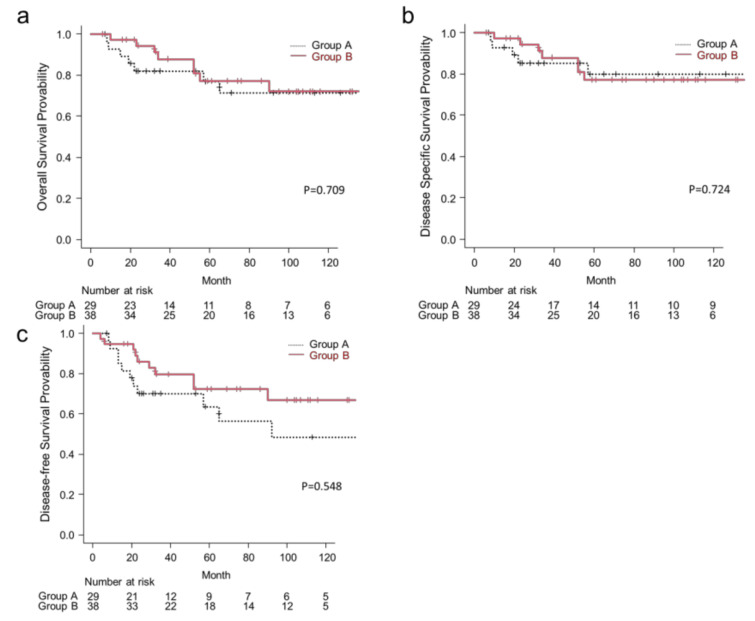
Kaplan–Meier curves of Group A and Group B ((**a**) OS; (**b**) DSS; (**c**) DFS). OS, overall survival; DSS, disease-specific survival; DFS, disease-free survival.

**Figure 2 medicina-57-00745-f002:**
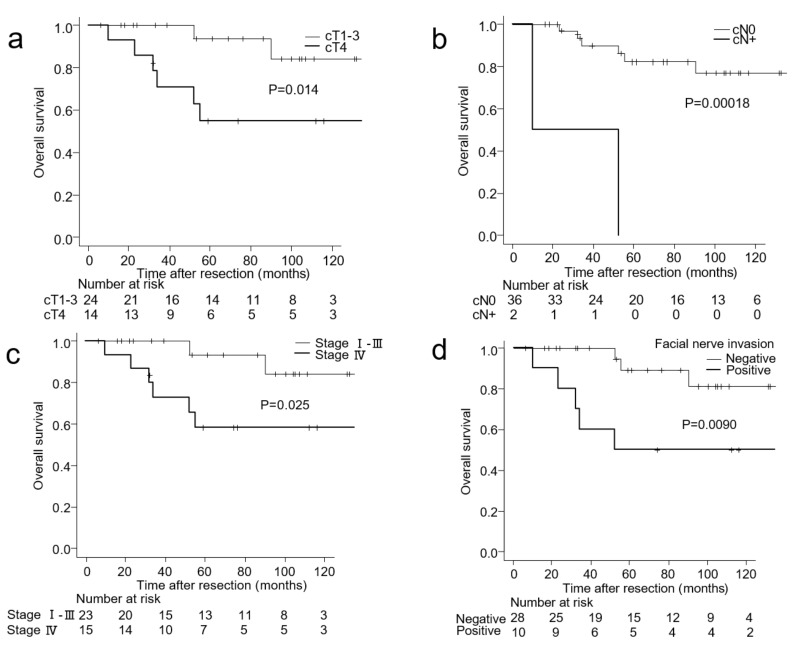
Kaplan–Meier curves of OS of Group B stratified by (**a**) clinical T stage, (**b**) clinical N stage, (**c**) disease stage, and (**d**) intraoperative facial nerve invasion.

**Table 1 medicina-57-00745-t001:** Patient characteristics (*n* = 67).

Variable	Group A (*n* = 29)	Group B (*n* = 38)	*p* Value
Median age (range), years	54 (15–74)	62 (20–89)	0.040 *
Sex			0.93
Male	18	22	
Female	11	16	
Preoperative facial nerve palsy			0.42
No	26	37	
Yes	3	1	
Clinical T classification			0.075
T1	2	9	
T2	15	9	
T3	4	6	
T4	8	14	
Clinical N classification			
N0	22	36	0.090
N1	3	0	
N2b	3	2	
N2c	1	0	
Disease stage			0.16
I	2	9	
II	12	8	
III	4	6	
IV	11	15	
Surgery			
Parotid resection			<0.001 *
Partial	4	31	
Total	25	7	
Facial nerve resection			0.0291 *
Preserved	17	26	
Partial	1	8	
Total	11	4	
Neck dissection			<0.001 *
Yes	12	2	
No	17	36	
Postoperative facial nerve palsy			0.15
No	10	21	
Yes	19	17	
Postoperative radiotherapy			0.75
Yes	14	21	
No	15	17	

*: *p* < 0.05.

**Table 2 medicina-57-00745-t002:** Histological grades and Histology.

Histological Grade	Histology	Group A (*n* = 29)	Group B (*n* = 38)	Total
Low		16	17	33
	Mucoepidermoid carcinoma, low grade	8	6	14
	Acinic cell carcinoma	8	7	15
	Epithelial-myoepithelial carcinoma	0	2	2
	Carcinoma ex-plemorphic adenoma, non- invasive	0	1	1
	Basal cell adenocarcinoma	0	1	1
Intermediate		3	1	4
	Adenocystic carcinoma, tubular type	1	1	2
	Adenocystic carcinoma, cribriform type	2	0	2
High		10	20	30
	Mucoepidermoid carcinoma, high grade	4	5	9
	Carcinoma ex-plemorphic adenoma, invasive	3	8	11
	Squamous cell carcinoma	1	3	4
	Adenocystic carcinoma, solid type	1	1	2
	Adenocarcinoma, NOS	0	1	1
	Salivary duct carcinoma	1	1	2
	Undifferentiated carcinoma	0	1	1

**Table 3 medicina-57-00745-t003:** Univariate analysis of clinicopathological factors for 5-year OS, DSS, and DFR.

(a) Group A						
Variable (Number)	OS (5 Years)	*p*	DSS (5 Years)	*p*	DFS (5 Years)	*p*
Age (years)		0.604		0.608		0.508
<60 (22)	73.9%		78.1%		56.2%	
60 or higher (7)	85.7%		85.7%		85.7%	
Sex		0.007 *		0.013 *		0.006 *
Male (18)	61.3%		65.7%		41.4%	
Female (11)	100%		100%		100%	
Clinical T classification		<0.001 *		<0.001 *		<0.001 *
T1–3 (24)	91.3%		95.5%		81.8%	
T4 (5)	0.0%		0.0%		0.0%	
Clinical N classification		0.208		0.043 *		0.009 *
N0 (22)	84.0%		88.2%		76.5%	
N+ (7)	53.6%		53.6%		19.0%	
Disease stage		<0.001 *		<0.001 *		<0.001 *
I–III (23)	95.5%		100.0%		85.7%	
IV (6)	0.0%		0.0%		0.0%	
Facial nerve invasion		0.348		0.075		0.17
No (17)	87.5%		93.3%		80.0%	
Yes (12)	55.6%		55.6%		37.5%	
Histology grade		0.147		0.026 *		0.274
Low/intermediate (19)	88.9%		94.4%		76.5%	
High (10)	34.3%		34.3%		28.6%	
Postoperative radiotherapy		0.788		0.218		0.96
Yes (14)	62.3%		62.3%		53.0%	
No (15)	85.7%		92.9%		69.2%	
**(b) Group B**						
**Variable (Number)**	**OS (5 Years)**	***p***	**DSS** **(5 Years)**	***p***	**DFS (5 Years)**	***p***
Age (years)		0.770		0.580		0.997
<60 (13)	73.3%		73.3%		75.5%	
60 or higher (25)	80.6%		86.3%		70.2%	
Sex		0.481		0.599		0.771
Male (22)	72.5%		78.5%		74.1%	
Female (16)	83.9%		83.9%		69.9%	
Clinical T classification		0.014 *		0.002 *		0.006 *
T1–3 (24)	93.8%		100.0%		87.5%	
T4 (14)	55.0%		55.0%		50.0%	
Clinical N classification		<0.001 *		<0.001 *		<0.001 *
N0 (36)	82.3%		85.8%		76.9%	
N+ (2)	0.0%		0.0%		0.0%	
Disease stage		0.025 *		0.004 *		0.013 *
I–III (23)	93.3%		100.0%		86.7%	
IV (15)	58.2%		58.2%		53.3%	
Facial nerve invasion		0.009 *		0.001 *		0.003 *
No (28)	89.2%		94.1%		85.7%	
Yes (10)	50.0%		50.0%		40.0%	
Histology grade		0.483		0.102		0.214
Low/intermediate (18)	84.6%		91.7%		86.6%	
High (20)	70.9%		70.9%		60.7%	
Postoperative radiotherapy		0.633		0.720		0.927
Yes (21)	81.9%		81.9%		72.6%	
No (17)	70.8%		77.9%		72.8%	

OS, overall survival; DSS, disease-specific survival; DFS, disease disease-free survival. *: *p* < 0.05.

**Table 4 medicina-57-00745-t004:** Recurrence cases.

No	Histology	cT	cN	cStage	Grade	PORT †	Recurrence Site and Management	Status at Last Follow-Up
**Group A**								
1	Acinic cell carcinoma	2	0	2	Low grade	−	LR: salvage surgery → controlled	58 months, alive
2	Acinic cell carcinoma	2	0	2	Low grade	−	LR, CLNR: salvage surgery → controlled DM: chemoradiation therapy	152 months, died
3	Mucoepidermoid carcinoma	4a	2b	4a	High grade	+	LR: palliative care	22 months, died
4	Mucoepidermoid carcinoma	2	1	3	Low grade	+	LR: salvage surgery → controlled	328 months, alive
5	Acinic cell carcinoma	2	0	2	Low grade	−	LR, CLNR: unknown	228 months, alive
6	Adenoid cystic carcinoma	4a	2b	4a	Intermediate grade	+	LR, CLNR, DM: unknown	8 months, died
7	Carcinoma ex-plemorphic adenoma	4a	0	4a	High grade	+	LR: unknown	57 months, died
8	Mucoepidermoid carcinoma	4a	2c	4a	High grade	−	LR, CLNR, DM: unknown	23 months, alive
9	Mucoepidermoid carcinoma	4a	2b	4a	High grade	−	LR, CLNR, DM: unknown	9 months, died
10	Mucoepidermoid carcinoma	2	0	2	Low grade	−	LR: unknown	272 months, alive
11	Squamous cell carcinoma	2	0	2	High grade	+	DM: unknown	19 months, died
**Group B**								
1	Mucoepidermoid carcinoma	4b	2b	4b	High grade	+	DM: salvage surgery → uncontrolled	52 months, died
2	Mucoepidermoid carcinoma	3	0	3	High grade	+	DM: salvage surgery → controlled	95 months, alive
3	Carcinoma ex-plemorphic adenoma	4a	2b	4a	High grade	−	CLNR: palliative care	10 months, died
4	Carcinoma ex-plemorphic adenoma	4a	0	4a	High grade	+	CLNR: salvage surgery → controlled	146 months, alive
5	Carcinoma ex-plemorphic adenoma	4a	0	4a	High grade	+	LR, CLNR: salvage surgery → controlledDM: palliative care	34 months, died
6	Acinic cell carcinoma	4b	0	4b	Low grade	−	CLNR: salvage surgery → controlled DM: palliative care	55 months, died

† PORT, postoperative radiotherapy. LR, local recurrence; CLNR, cervical lymph node recurrence; DM, distant metastasis.

## Data Availability

Data will be made available upon request.

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
