# Peer review of "Impact of Changing Surgical Strategies on Clinical Outcomes in Patients with Parotid Carcinoma: A 53-Year Single-Institution Experience"

_medicina, 2021, doi:10.3390/medicina57080745_

Round 1

Reviewer 1 Report

The authors present an interesting study on the management of parotid carcinoma. Moreover, outcomes as they are related to the altered surgical strategy. The improvement in the final function of the facial nerve are outlined. The paper is of interest for those involved in the management of parotid carcinoma.

Author Response

We appreciate your review of our manuscript. The comments of the reviewers have been helpful in allowing us to revise our manuscript. Our responses to the reviewers’ comments are as the attachment.

Reviewer 2 Report

This study examines a 52 year experience with parotid carcinoma, specifically comparing an earlier approach (total parotidectomy +/- neck dissection) with partial parotidectomy.  Although the safety and efficacy of partial parotidectomy with facial nerve preservation has been well established, this paper does add the overall experience with this approach.  In that regard, the paper is not novel but reinforces what other studies have shown and would likely be of interest to readers performing salivary gland surgery.

The authors describe performing neck dissection but do not elaborate on type - selective, modified radical, radical?  A selective level 2-3 neck dissection does not introduce significant morbidity.  The authors may want to discuss this.

Given that this study covers a 50+ period of time, there have been notable advances in imaging technology as well as radiation techniques (e.g. IMRT) that could also impact staging of the cancers treated as well as outcomes.  For example, recurrence rates in group A were higher than group B - although surgical technique/experience as the authors mention could explain this difference but could differences in radiation technique also play a role?  

Page 3, line 109 - I believe for group A the authors meant to say "total parotidectomy" instead of "partial parotidectomy"

Page 4, line 145 - did the authors mean "rate" as opposed to "late"?

Page 6, line 190 - This sentence is confusing

Page 11 lines 242-245 and lines 253-256:  I am having trouble understanding these sentences:

"Therefore, the incidence of postoperative facial nerve palsy was high until 2002: 47.1% among patients with facial nerve preservation and 61.5% among patients without preoperative facial nerve palsy."

and

"Thus, the incidence rate of facial nerve palsy in the current surgical strategy decreased to 26.9% among patients with facial nerve preservation and 43.2% among patients without preoperative facial nerve palsy,"

Did the patients with "facial nerve preservation" have a facial palsy preoperatively?  Where did these percentages come from?  They are not in the results.  Please correct this.

Author Response

We appreciate your review of our manuscript. The comments of the reviewers have been helpful in allowing us to revise our manuscript. Our responses to your comments are as the attachment.

This manuscript is a resubmission of an earlier submission. The following is a list of the peer review reports and author responses from that submission.

Round 1

Reviewer 1 Report

I have read the manuscript related to parotid cancer in considerable detail. My major criticism: it is a small series of patients over an extended period of time suggesting very few cases every year, no new findings. The treatment philosophy of partial versus total parotidectomy has been a subject of debate over the years, however, the concept of malignant surgical resection and oncologic satisfactory procedure is well known both in surgical oncology and also in parotid surgery. There has been considerable shift performing limited parotid surgery in selected patients. The idea is to resect all gross tumor with satisfactory margin. This has been a principle in the past and also recently.  The idea of preserving the functioning facial nerve has been the goal in parotid cancer surgery.  A substantial number of T3 and T4 parotid cancers will require postoperative radiation therapy and interpreting the data of surgery may be somewhat difficult in this group. The authors have a small series of patients and more importantly there are a variety of different histologies and T stages.  So, the manuscript is difficult to interpret or make any conclusions. The authors have also used their own series of patients treated previously, however, there is no good data in this manuscript related to their presumption.  I don't think this manuscript will add much to the existing literature. It is maybe rather something to have a resident present at a national meeting than really deserving to be published.

Reviewer 2 Report

Although much has been written concerning the outcomes of surgery of parotid carcinoma, the present study is the first one to compare postoperative outcomes before and after the implementation of
distinctive alterations in the surgical  policy  at a single institution. the authors should be congratulated for their endeavor. The paper is well organized, the authors use appropriate statistics, results are clearly presented and the discussion is extensive with an up-to-date reference list. The paper should be read by all surgeons interested in the management of parotid carcinoma.